# The Influence of Innovation in Tangible and Intangible Resource Allocation: A Qualitative Multi Case Study

**Rui Silva** [1,]* and **Cidália Oliveira** [2]

1   CETRAD (Centre for Transdisciplinary Development Studies), University of Trás-os-Montes e Alto Douro, 5001-801 Vila Real, Portugal

2   NIPE (Centre for Research in Economics and Management), University of Minho, 4710-057 Braga, Portugal; cidalia.oliveira@eeg.uminho.pt

*   Correspondence: ruisilva@utad.pt

**Abstract:** Considering the current turbulent macroeconomic environment, the aim of this research is to explore the influence of innovation in tangible and intangible resource allocation. The literature underlines that organizations are facing a revolution in their business processes. As such, there is a need to understand the value of knowledge resources and to identify ways to manage them. This paper explores the field of resource allocation, namely dynamic capabilities, and highlights the importance of monitoring intangible resources. This research has three specific contributions. The first contribution provides a comprehensive picture of what has occurred in the field of tangible and intangible resource allocation, such as intellectual capital and its importance towards organizational performance. Secondly, it offers evidence about the actual need for performance measurement tools that foster intangible resource monitoring. Organizations devote special attention to market demands which consequently lead managers to adapt their strategies in areas concerning resource allocation. Given this importance, this research, comprising major innovative organizations in Portugal from diverse activity sectors, provides new insights and stresses the importance of tools to follow the overall performance of resource allocation. Managers of innovative organizations recognize the very powerful features of the Balanced Scorecard (BSC) in monitoring and linking strategic resources of both tangible and intangible natures. Thirdly, this research, with a view to enrich the field of intangible natures, points out some aspects for future research areas, bearing in mind the relevance of this research area confirmed by managers of the major innovative organizations. Thus, it provides prominent information for both academia and innovative organizations.

**Keywords:** innovation; intangible resources; learning and growth; intellectual capital; balanced scorecard

## 1. Introduction

Currently, the environment in which organizations operate and establish commercial relationships is characterized as a challenging scenario, which leads managers to resort to robust business strategies. Organizations are required to be able to respond, in a very short time, at the lowest cost. In addition to this requirement, they must remain focused on efficient execution, becoming increasingly dynamic, based on the impetus of globalization. This era of globalization calls for new approaches, bearing in mind the relevance of intangible resources, like learning and growth and intellectual capital, in which the traditional strategy is no longer satisfactory.

In this sense, in order to allow for the flexibility required by the market, and to foster organizational performance, the theory of dynamic capabilities emerged as a complement to the theory of Resource

Based View (RBV). This theory of dynamic capabilities intends to correspond to the need to adapt resources in turbulent and increasingly competitive environments. Thus, through the dynamic configuration of resources and competencies, companies assume a more competitive role. Consequently, managers confirm a need to ensure a better control [1–3]. The organizational knowledge of resources and skills is the key factor in making knowledge valuable and sustainable [4].

In economic and social contexts, decisions must be made quickly, knowing that if decisions are postponed, certain opportunities may quickly be missed [5].

Recent studies try to understand the dimensions of dynamic capabilities that influence company's performance [6] and others devote attention to interactive profit planning based on dynamic capabilities [7]. We especially highlight the research of Giniuniene & Jurksiene [8] that fosters the relationship between dynamic capabilities, organizational learning and innovations' impact on company's performance. Giniuniene and Jurksiene [8] performed this conceptual research to promote further empirical studies.

Bearing in mind the call for empirical research stated by Giniuniene and Jurksiene [8] and the identified gap in the literature, our paper tries to fulfil new insights. Namely, this research recognizes the influence of innovation in tangible and intangible resource allocation and furthermore stresses the importance of performance measurement tools for a more efficient monitoring of tangible and intangible resource allocation.

Considering the relevance that is currently devoted to resource allocation and its performance measurement, the following Proposition was put forward:

1. Explore the influence of Innovation in tangible and intangible resource allocation
2. Recognize the importance of performance measurement tools for a more efficient monitoring tangible and intangible resource allocation.

The literature underlines that organizations are facing a revolution in their business processes, therefore this study contributes to the area of Management Control in an organization whose activities are related to innovation. Managers that trust performance tools are able to monitor and to reallocate their resources in a precise and detailed way.

This work is structured in several sections that will be presented below. After the introduction of the themes in the analysis, we present a literature review on innovation, markets and tangible and intangible resource allocation. To follow, we present the qualitative research methodology used and the results of the research carried out. Finally, we present a discussion and conclusions of the work, as well as its limitations and future lines of research.

## 2. Literature Review

### 2.1. The Relevance of Knowledge for Dynamic Capabilities

In the past, the competitive advantage lay in the successful exploitation of economies of scale, unique technologies, or in the dominance of certain markets or supply chains. However, as time went by, it became clearer that competitive advantages are achieved based on primarily intangible resources. The key members of a given market are those who have access to unique resources, which are difficult to imitate, which is why—in this sense—the technological boom was said to be of enormous importance. This innovative way of disseminating and sharing information, knowledge and organizational practices has contributed considerably to the evolution of organizational models [9].

Although these resources in general are identical between companies in the same industry, it is the different allocation and adaptation of resources that allows for the distinction between companies. The resources have to be characterized by the organization so that they are not easily transferable. Based on this need, it is imperative that companies devote special attention to the constant fluctuations of the market in order to continually keep their resources up to date. The value of a given capacity can be calculated in isolation, or added to a given resource [10,11]. Dynamic capabilities are considered

capabilities capable of creating, expanding and modifying resources according to the needs of the environment [12,13]. In this sense, there are three mechanisms that inhibit the imitation of resources, namely ownership of rights, learning and development costs, as well as causal ambiguity [14].

Briefly, the RBV theory can be characterized as a theoretical model to understand how a competitive advantage is achieved, as well as how it can be sustained [15]. Human capital is considered to be one of the main resources in organizational adaptation, so that the valuation and the organizational learning process are vital for development [16].

It is worth emphasizing that the more that certain resources are analyzed in detail, the easier it will be to adapt them to new realities and, to this end, the real time factor must be incorporated. Based on timely and detailed information, intra-organizational, inter-process and extra-organizational communication is made possible.

To meet this need for more dynamic capacities, organizations have been increasingly investing in qualified employees, as well as in a closer connection with business partners in order to be able to understand what really generates value for organizations. Often, companies have, as a main concern, the concentration of resources and necessary competences only for the short term, but dynamic capabilities must be considered in the long term. Thus, they can ensure that companies are competitive and innovative, especially in increasingly global and dynamic markets. If organizations strategically opt for a reduced investment, this will result in less innovation; on the other hand, if they opt for excessive investment, they will not have enough time to obtain the necessary return, due to constant market fluctuations [15].

Through dynamic capacity management, organizations show the ability to detect and take advantage of new opportunities [16], so dynamic capabilities should be seen in a bipartite analysis, that is, in the internal and external capacity of the organization, related with the environment [17]. All organizations must be managed and organized efficiently, as referred by [18], as the organization's objective is to simplify the coordination of processes and the respective allocation of resources. For organizations to reach opportunities at the heart of the market, resources are assumed to be crucial; to this end, it is imperative that they work at a lower cost while maintaining a higher level of quality. Beyond the perspective of the specific resources of a specific organization, as mentioned by [19], as well as [15], there are certain resources that can be transversal to the same activity, not specific to an organization. Evolution, based on dynamic capabilities, is achieved through the alteration, adaptation and reconfiguration of specific internal competencies, as well as external factors [15]. Based on empirical studies, [20] confirms that both for knowledge and for internal and external processes, the efficient configuration of dynamic capabilities is imperative. He also stresses that knowledge based on experience is very valuable, existing in this way as effectively a valuable and fundamental source of progress for organizations. It is precisely this experience that makes dynamic capabilities, focused on adapting capabilities dynamically to the market, a reality.

In general, all organizations benefit as long as they have an information system, supported by a network system, through which they can maximize and share knowledge [21]. As a practical example, the Toyota case can be considered, in which the implementation of knowledge sharing and collaboration between partners along the same supply chain showed enormous benefits, as all partners were considered members of the same network of knowledge and sharing [21].

This organizational effectiveness is achieved based on a standardization of knowledge, adapting the routines according to the demands of the market. The adaptation of operational routines to dynamic capacities leads to efficiency [22,23]. The contribution of the authors Zollo and Winter [23] was clearly relevant, as they clarified ways of accumulating and articulating organizational experiences in order to sustain their organizational activities. According to them, knowledge must follow certain steps: the accumulation of knowledge, articulation of knowledge, codification of knowledge. For this, they need to have resources with dynamic configurations in order to exceed the expectations of their customers, thus becoming well publicized in the market. It is through knowledge that dynamic capabilities, as well as operational capabilities, leverage their robustness [23]. According to Lavie [24],

the reconfiguration of resources may be through substitution, evolution and transformation. Dynamic capabilities are distinguished from substitution capabilities, as they do not merely replace previous, already obsolete resources; they choose to reconfigure them [24].

In the perspective of Zollo and Winter [23], dynamic capabilities consist of learned and stable patterns, which form a collective activity through which the organization systematically generates and adapts its routines, in favor of internal efficiency. However, knowledge transfer is a difficult process to be performed, therefore it is required that managers have bivalent capacities, namely in cognitive and human capital areas, in order to generate positive return [25].

Certain performance measures based on resources of an absorptive nature are not clearly defined throughout the literature, so Zahra, Sapienza, Harry and Davidsson [26] pointed out a new definition. According to this new point of view, experience, customer orientation, as well as knowledge and diversity are considered advantages for organizational development. Ambidextrous policy-making organizations are generally composed of several subunits, based on cultural policies, inter-unit and organizational procedures, in which market segmentation makes the whole process easier [27].

Dynamic capabilities can be defined by three different dimensions (detection, integration and reconfiguration capabilities). In turn, they promote different types of innovation that improve the company's performance [6].

In order to provide an updated overview of the relevant literature regarding the relevance of dynamic capabilities, the Table 1 found below was composed:

**Table 1.** Relevant Literature Review—Dynamic Capabilities.

| Author | Year | Aim | Methodology | Main Conclusions |
|--------|------|-----|-------------|------------------|
| Salunke, Weerawardena, & McColl-Kennedy [28] | 2019 | Exploit how B2B service companies organize and manage knowledge in order to achieve competitive advantage | B2B organizations related to projects of Australia and USA | The knowledge acquired through external and internal sources, must be integrated with the existing knowledge, to provide innovative services that correspond to the needs of the customers. |
| Zhou, Zhou, Feng, and Jiang [6] | 2019 | Understand different dimensions of dynamic capabilities that influence company's performance | 204 Chinese firms, SEM | There are mechanisms dimensions (detection, integration and reconfiguration capabilities) of dynamic capabilities which influence the company's performance. |
| Peters, Gudergan and Booth [7] | 2019 | Interactive profit planning based on dynamic capabilities. | Interview of 331 Australian firms | Greater market turbulence strengthens the effect in interactive resources and greater market turbulence reinforces the positive effect of flexibility values. |
| Kien, Vinh, Minh, and Vo [29] | 2020 | Political policy's effect on innovative activities | 2600 companies in Vietnam, | Innovative activities by itself, in small and media companies, do not promote corporate performance. |
| Mikalef, Krogstie, Pappas and Pavlou [30] | 2020 | Competitive Power | Survey data from 202 chief information officers and IT managers working in Norwegian firms (SEM modeling) | Dynamic capabilities have positive effect and are significant in two types of operational resources: marketing and technological resources. |
| Giniuniene and Jurksiene [8] | 2015 | Relationship between dynamic capabilities, organizational learning and innovations' impact on company's performance | Conceptual research paper | Theoretical model for establishing empirical tests. |
| Fainshmidt, Wenger, Pezeshkan, & Mallon [31] | 2019 | The relationship of dynamic capabilities and competitive advantage is dependent on the strategic fit between organizational and environmental factors | Comparative qualitative analysis | Strategic adjustment between organizational and environmental factors, require a rigorous assessment of configurational dynamic resources. |

Bearing in mind the relevance of knowledge and the impact on resource allocation, literature reveals the need to understand the impact of market turbulence and technological intensity, which will be explores along the next section.

## 2.2. Market Turbulence and Technological Intensity

Organizations are inserted in an economic and social environment, which has a strong influence on them, because, in fact, organizations have to adapt to the demands of the external environment, in order to remain competitive. In the past, several market-leading organizations have seen their position in decline, as managers have not been able to see further, innovate, adapt their resources and explore their experiences. This organizational myopia has led several successful companies to regress, due to their "myopia" of adapting to the market [32].

Nowadays, organizations are constantly adapting to market demands, which lead them to be overhead-intensive [33]. Organizations must take an active role in adapting to the environment, based on their resources and skills [34]. The dynamic advantage in turbulent markets is only attainable due to the ability of organizations to adapt continuously to changes in the environment, customer requests and available technology [35]. The center of organizational attention is focused on adapting quickly to environmental requirements and the consequent need to create and renew resources, as well as reconfiguring the range of existing resources [36,37]. Based on dynamic capabilities, organizations are allowed to reconfigure themselves according to the market, however due to this need for constant adaptation, competitive advantage is not fully attainable [15,35]. Capabilities differ precisely because of the speed with which they adjust, since it is exactly this faculty that differentiates them from ordinary capacities [38].

In turn, these changes and requirements of the environment, exogenous to organizations, may reflect significant opportunities for organizations that meet the requirements, even allowing for the achievement of long-term competitive advantage. For this advantage to be achieved, it is imperative that organizational capacities are sustained in endogenous organizational relationships, namely substantive capacities [32].

Especially in turbulent markets, with constant changes, capacities show their strengths demanding that the most common capacities, called nouns, adapt, and reconfigure according to the market [13]. In the face of extreme changes, organizations have to pay special attention in order to remain competitive. Organizational managers must maintain the main focus on substantive capabilities in order to be able to change them in accordance with market requirements, turning them, through dynamic configurations, into dynamic capabilities, thus allowing the sustainability of the company in a wider field of action. There are monthly possibilities for small organizations, or even start-ups, which see their market as their first opportunity based on their resources. These opportunities generally prevail in highly technological markets. Likewise, internationalization can be considered a bet, based on dynamic capabilities, so even embryonic companies can be successful in internationalization. For this, it is necessary that companies configure their resources dynamically, and allow the exchange of information with managers from other units in international territories. This bet should also be reflected in universities, promoting exchanges as these are fruitful to international dynamism and flexibility.

In turbulent environments, dynamic capabilities characterize simple, highly experimental, fragile and unpredictable processes due to their surroundings [15]. It is based on these competencies, namely dynamic capabilities, that organizations see the possibility of creating new products and processes according to the demands of the environment. In order to be able to satisfy these organizational needs, it is extremely important that organizations bet on areas such as research and development. Organizations essentially want technological knowledge to be promoted in order to achieve economies of scale, which, however, does not mean that resources have to be produced by a single company.

Kien et al. [29] investigate the joint effects of innovation policies and strategies on corporate profitability.

Analyzing more than 2600 companies in Vietnam, during the period from 2005 to 2015, the results reveal that the political policy mitigated the inefficiency of the innovative activities focused on the performance of the organization, allowing to positively promote the innovative capacity and the most significant profits.

However, innovative activities alone, in small and in media companies, are not conducive to corporate performance. The same is true for the political connection, as it increases the number of contacts or the political time of interaction, or the value of the company will be reduced [29].

Peters, Gudergan, & Booth [7] empirically investigate the concept of interactive profit planning supported by dynamic experiments. Interactive Profit Planning System (PPS) resources and routines are the upper and middle levels for knowledge creation, detection, apprehension and reconfiguration or business model. Peters et al. [7] confirms that greater market turbulence strengthens the effect of PPS interactive features and greater market turmoil reinforces the positive effect of flexibility values. Therefore, the interactive features of PPS behave according to the principles of dynamic capabilities [7].

As the recent literature highlights the request for competitive power of dynamic capabilities, in order to correspond to market's request, specific analysis to explore the competitive power will be explored in the following section.

### 2.3. Competite Power of Dynamic Capabilities

Based on this differentiation, organizations that are more attentive to changing needs and that are able to adjust more quickly, based on product innovation, will be the most competitive [19]. For competitiveness to be maintained, organizations must focus on market orientation, management knowledge and customer management relationship (CRM) as these aspects are crucial for the creation of value. For an organization's resources and competencies to be identified as dynamic capabilities, they will have to be based on standardized behaviors [12,13]. As an example of dynamic capabilities, organizational alliances and new product development can be affirmed, altering the ways in which organizations sustain themselves [15]. Dynamic capabilities are characterized by very specific areas and activities, as referred to by Helfat et al. [12] and Winter [13].

Several managers stressed the need to have tools to track and measure the influence of organization communication, as—due to its intangible nature—they felt the difficulty to link and measure its influence. In the words, the relevance of communication is known in different organizational fields as, without communication, no task would be fulfilled successfully due to the lack of link or relationship to other organizational processes or actions. By means of the Balanced Scorecard (BSC), managers are able to define and understand causal links between indicators that, consequently, will enable organizational communication.

It is well known that organizations that have the capacity to exploit their internal forces diminish their limitations, reducing external threats and have the possibility to explore the opportunities of the environment [38,39]. Consequently, based on competitiveness, they will find that organizational success is more easily achieved [40]. Knowing that the oscillations are constant and very extreme, it becomes necessary that they are controlled and guided [41]. For that, organizations are considered open media, which receive and transmit information, having to adapt to extremely dynamic environments [42]. Communication regarding decision making is extremely relevant [43].

Commercial practices demand that they adapt and proceed according to the requirements, adapting products and services to maintain competitiveness. The need for managers to transform and adapt their organizations in a sui generis way is evident, without depending on or creating organizational routines, but rather configuring them according to the intended requirements [44]. Given that the external forces of the environment are uncontrollable, there is a need for organizations to remain alert and flexible. In view of this need for constant adaptation, it is imperative that knowledge accompanies this evolution, as previous knowledge is considered insufficient [45–47]. In order to keep knowledge alive, organizations must pay special attention to the promotion and development of new knowledge as the environment directly influences the performance of organizations [48]. Precisely in turulent markets, the need to establish, maintain and develop commercial relations is increasingly prominent, so the strategic vision between partnerships can be fostered and based on a Business Relationship Process Management (BRPM) in an aggregate way [49]. This plan consists of an interconnection competence between the various business partners, dealing with the inherent

threats and opportunities, knowing that Business to Business (B2B) competences currently have to be extended to the scale of the supply chain. The ability to implement new routines in favor of product development is called substantive capacity, while the ability to change capacities is called dynamic capacity. The process that guides and monitors dynamic capabilities is a time-consuming, rigorous, dynamic and demanding process [50].

To explore technological and market knowledge, partnerships and commercial partners collaborate and propose new alternatives as internal resources go beyond the capacity to produce, since they involve tangible and intangible assets [19]. Through dynamic capabilities, it is possible to have a competitive advantage in partner companies because, through the aggregation of resources, it is possible to sustain dynamic capabilities [51]. The importance of dynamic capabilities is not only directed at managers, but also at entrepreneurs, as they have to be able to look ahead to be the first to identify an opportunity within a specific market, framed with the intended positioning.

Salunke, Weerawardena and McColl-Kennedy [28] concentrated on understanding how B2B service companies organize and manage knowledge in order to achieve competitive advantage, addressing the role of the antecedents of integrating knowledge and capacity in the competitive advantage fostered by innovation.

Recent literature addresses a central issue with Big Data, without understanding how they can help achieve competitive reach. Mikalef, Krogstie, Pappas, & Pavlou [30] developed an investigation about dynamic resources, examining an indirect relationship between a company's big data analysis capacity (BDAC) and competitive performance. This study showed that a strong BDAC can promote a competitive advantage. Despite this being indirect, it is recognized, as the use of dynamic capabilities has a positive effect and they are significant in two types of operational resources: marketing and technological resources.

Based on the literature that highlights the relevance of having adapted and updated resources, these need to be, on one side, fostered by constant learning, and on the other side, be monitored in order to follow its evolution. Considering this aim, the following section devotes special attention and highlights the research related to the monitoring of learning and intellectual capital.

## 2.4. Monitoring Learning and Growth and Intellectual Capital

Nowadays, organizations are facing the effects of a digital revolution [52] which has led to an update of management tools [53].

Several managers implemented the Balanced Scorecard not only to measure performance, since it became part of budgeting and planning activities, but mainly because ambitious goals related to capital and investment are possible [3]. It also appears that certain organizations have chosen to implement the BSC as a strategic management system, monitoring both past and future performance [54]. This tool adds value through relevant and balanced information that is concisely conveyed to managers [55]. One of the advantages of having the BSC successfully implemented is the ability of managers to adapt it according to the specific needs of each organization [56]. The tool is now considered a fundamental work base [57], with a tendency to be increasingly implemented. However, in order for the BSC to be able to demonstrate its added value and for its expected benefits to be achieved, it is necessary to initially determine whether the conditions for their implementation are met [58]. Based on the BSC, organizations make their information more transparent, enhancing relationships with their members (suppliers, customers and other entities), and also make it possible to provide more complex analyses and data. Additionally, this tool may bring up the identification of inadequate strategic decisions, further enhancing feedback from managers [58].

Managers who define the indicators and monitor the entire process starting at implementation, namely rolling out, allow the BSC to go beyond a causal link process. On average, the rolling out of this tool, after defining the surrounding strategy, will take eight to twelve weeks. However, one of the most complex preliminary tasks resides in the selection of the work team, in its structuring and in the respective financing. The team assigned to monitor the implementation must define strategic

plans subdivided into financial plans, customer segmentation plans, human resource plans and quality plans. This comes along with other relevant information, as this tool will have an aggregate function, acting in the areas of control, information, communication and learning. In short, this management tool measures, controls, informs, communicates and encourages structural changes [59].

The environment of the BSC, in view of strategic management, highlights communication and liaison, business planning, translation of vision, as well as response and learning. Among these factors, customer satisfaction can be highlighted as the central focus of the organizational strategy, as customer satisfaction is essential to achieve the objectives [3].

The Learning and Development Perspective is characterized by measuring the learning and development of employees, who promote organizational growth. The forecast of growth, research and development of new products, as well as the development of human resources, are integrated in this perspective [3]. This perspective identifies the objectives and indicators that support and allow the evolution of skills to foster the development of the organization. In other words, it puts into practice what is identified by other perspectives, seeking to provide answers to questions such as: "where?", and "what is it like?". Thus, this perspective reflects the necessary measures for skills to be developed even if other perspectives are needed. This perspective is considered the lever of the others, as it makes the results of the first three perspectives viable. However, for this, it is crucial to invest in the present so that, in the future, the infrastructures, skills and resources correspond to the demands of the market, making the training of employees, information systems, as well as motivation, all the more important [3].

In this way, the perspective of learning and development consists mainly of the capacity that the organization has in learning, adapting and growing. In this sense, the resources that the organization allocates to research and development, especially human resources, are extremely relevant [60]. In relation to the perspective of learning and development, there is a concern with codifying knowledge, which may lead to tangible benefits in organizations directed to the public (final consumer), with heterogeneous characteristics and important routines [61]. The ability of organizations to disseminate new ideas and new products translates directly into added value for them, as new products foster customers' interest. This position at the forefront attracts new customers and enables an increase in profitability [59].

Literature confirms the importance of tangible assets, such as property, plant and equipment, but it is known that these have been overcome by new assets of intangible natures, such as knowledge-based ones. Bearing this shift in mind, the managing of these assets, namely brands, trade secrets, distribution channels or specific competences has become a critical aim, as managers are willing to measure and monitor these intangible assets [62]. Leite and Porsse have combined two areas related to strategy, namely the strategic position school and the resource-based theory due to the uncertainty ground and the consequent strategic change which promotes organizational learning and knowledge improvement [63]. Intellectual capital has increasingly become the focus of attention, as a substantial amount of research has been developed in this field. Nevertheless, rigorous empirical analyses need to be performed to confirm results and guide the measurement of intellectual capital [64]. Indeed, intellectual capital has gained the attention of researchers in an isolated and—especially—interlinked way to other connected research areas, like knowledge-based view and the need to measure organizational knowledge assets. The relevance of measuring knowledge assets is due to the need to define strategic key performance indicators [65].

The actual dynamic of markets and the requirement to adapt based on information and knowledge fosters the growth of intellectual capital. Considering the value of know-how, the need to define new metrics, to enable the report of knowledge, raised up [66].

Several researchers have devoted attention to intellectual capital, but just a few have dedicated attention to the organizational understanding, namely the organization's aim [67]. Inside organizations, the measurement of intellectual capital has to be improved as, in general, organizations do not have a measurement tool to monitor intellectual capital since they are measuring intellectual capital with

pure financial instruments [68]. Besides the reflected improvements that are still open to be performed related to the measurement of intellectual capital, it is decisive to confirm that intellectual capital plays a relevant role in knowledge management and intangible assets are relevant factors [69]. The literature recognizes several difficulties in empirically measuring the concept of dynamic capabilities and their relationship with other variables [8].The conceptual model of dynamic capabilities and their interdependence with other significant variables may shed light on the question of why some companies achieve competitive advantages in volatile environments, while others are unable to achieve success [8].

Recent literature indicates a relationship between dynamics and participation, which, however, can be affected by organizational and environmental factors [30].

Fainshmidt et al. [31] developed a theoretical model, showing how they dynamically trigger a competitive advantage when they support each other and develop an appropriate strategy to use in the environment.

In stable environments such as dynamic roles, they play an effective role in providing low cost support and guidance. Fainshmidt et al. [31] stated that a relationship of dynamic wear and competitive advantage depends on the strategic adjustment between organizational and environmental factors, requiring a rigorous assessment of configurable dynamic resources.

## 3. Methodology and Participants

### 3.1. Method

Considering the aim of this research, namely to gain more in-depth knowledge of demands related to innovation and its consequent resource allocation, the following propositions have been defined:

1.  Explore the influence of Innovation in tangible and intangible resource allocation
2.  Recognize the importance of performance measurement tools for a more efficient monitoring tangible and intangible resource allocation.

To this end, qualitative research, in the shape of interviews that were carried out with five Managers of major innovative organizations, was considered relevant in that it would allow for further analysis and comparison with the literature. Given the exploratory nature and the starting point of this study, a case study method was used to obtain a more in-depth understanding [70] and to explore the influence of Innovation in tangible and intangible resource allocation. The literature highlights the fact that organizations are facing a revolution in their business processes. When a scientific field still needs more exploration and interpretation and is still underexplored, with a substantial shortage of preliminary research on it, exploratory case studies are recommended [70]. In this sense, methodological procedures are essential in a case study, which implies that the research questions are properly supported by the literature review [70]. This study focuses on five major innovative organizations from different areas. These companies were selected in order to compose a different range of organizations from different sectors to ensure a representative view, namely financial service, food sector, automotive industry, civil construction, service providers. Based on this diverse range of organizations, we intend to contribute to a wider knowledge as it is not limited just to one sector of activity. It is our understanding that having a broad range of organizations will allow for a wider and more complete understanding, bearing in mind that the aim of this research is devoted to understanding the relevance of tangible and intangible resource allocation, such as intellectual capital. We consider that a key factor for innovative success may be linked to the capacity to efficiently allocate resources and promote intellectual capital.

In the flowchart below we can better understand the methodology used and the various steps of analysis of the qualitative data obtained in the interviews (Figure 1).

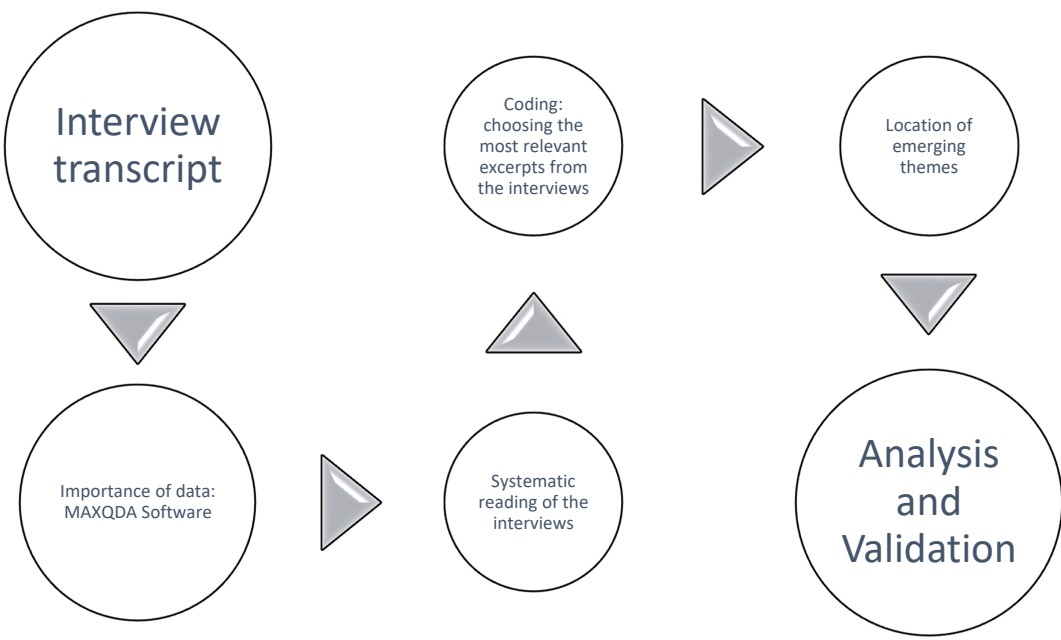

**Figure 1.** Description of the data analytical process.

*3.2. Participants*

Although there are several manuals of methodological procedures for research in social sciences, the case study follows its own methodological perspective and there is not only one method for data collection [71]. In this sense, the sources of empirical evidence used in this study, in an exploratory case, were semi-structured personal interviews (primary sources). The semi-structured interview is one of the most used methods in qualitative research, aiming at a complete understanding of a given social phenomenon, based on the personal experiences of the interviewees [72].

For this case study, five interviews were conducted, based on questions that resulted from the literature review (Appendix A, to the hereafter named P1, P2, P3, P4 and P5 who, despite authorizing the recording of the interviews did not authorize their identification in this research). Table 2 shows the profile of the participants.

**Table 2.** Interviewee profile.

| Code | Sex | Age | Years of Experience | Qualifications | Occupation | Interview Duration |
|------|-----|-----|---------------------|----------------|------------|--------------------|
| P1 | Male | 39 years | 14 | Master | Member Board Directors | 65 min |
| P2 | Female | 46 years | 22 | MBA | Controlling Manager | 60 min |
| P3 | Male | 45 years | 23 | Bachelor | Development Manager | 57 min |
| P4 | Male | 54 years | 32 | Bachelor | Administrative and Fin Manager | 72 min |
| P5 | Male | 50 years | 27 | PhD | General Manager | 75 min |

*3.3. Data Analysis*

The content analysis of the interviews was carried out using the MAXQDA18 software, allowing the analysis of the content of the interviews to be made in a more efficient way, with greater emphasis on emerging themes and units of meaning. In order to obtain greater consistency, and the maximum verisimilitude of the interpretations, a circular logic of conjecture and validation was adopted that governs the hermeneutic principle [73]. Then, in order to obtain a greater congruence between the proposals of emerging themes and units of meaning, a new content analysis of the interviews was carried out.

Using MAXQDA18, the most relevant excerpts from the interviews (meaning units) were coded according to the theme under analysis, later converting them into relevant expressions (emerging themes). It was thus possible to interpret the results in order to establish hierarchies and assumptions between the codifications. It should be noted that the fact that the results come from a process of interpreting the interviews may entail some subjectivity.

*3.4. Cohesion between Reports*

By means of qualitative analysis software, the cohesion between the participants' reports was analyzed through the correlation between the units of meaning and the emerging themes. According to the software outputs (Table 3), there is a strong correlation in interval [0.75 to 1] between the various contributions of respondents to this research study, and the closer the correlation is to 1, the greater the contribution of the interviews to the understanding of the phenomenon under analysis [74].

**Table 3.** Similarity Matrix.

| Participants | P1 | P2 | P3 | P4 | P5 |
|---|---|---|---|---|---|
| P1 | 1.00 | 0.75 | 0.88 | 0.88 | 0.88 |
| P2 | 0.75 | 1.00 | 0.63 | 0.88 | 0.75 |
| P3 | 0.88 | 0.63 | 1.00 | 0.75 | 0.88 |
| P4 | 0.88 | 0.88 | 0.75 | 1.00 | 0.75 |
| P5 | 0.88 | 0.75 | 0.88 | 0.75 | 1.00 |

## 4. Results and Discussion

The main objective of this study was to analyze the individual testimony of five interviews, so as to understand their position regarding the importance of resources and intellectual capital in their innovative performance. As these five organizations are among the most innovative organizations in Portugal, our research aim is to understand the relevance that they attribute to the allocation of both tangible and intangible resources, such as intellectual capital. To our understanding, a key factor for achieving innovative success may be related to their capacity to allocate efficiently resources and foster intellectual capital.

The result of the content analysis of the semi-structured interviews showed the main emerging categories that arise from the content analysis, characterizing the interviews conducted according to their occurrence (Figure 2). The thicker arrows mean a greater occurrence of these themes throughout the interview.

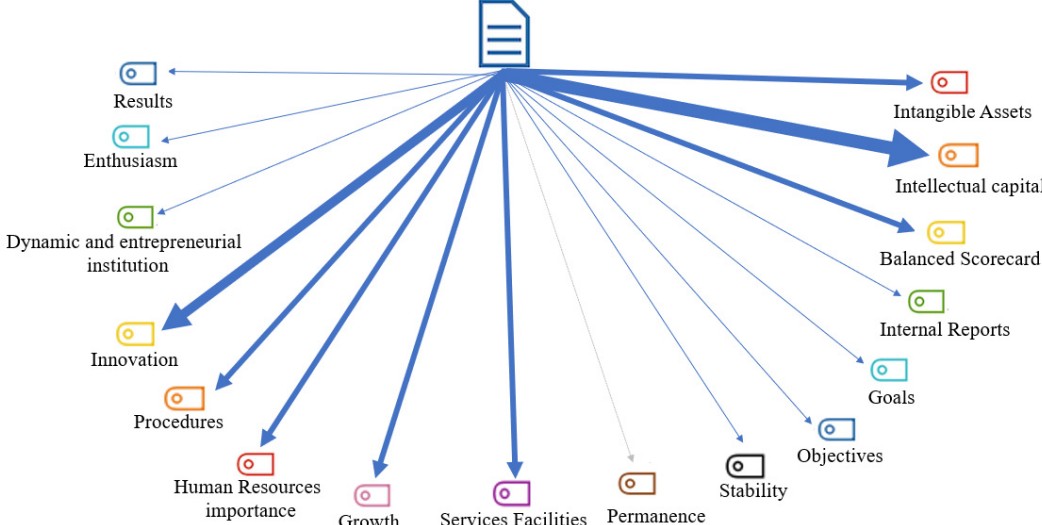

**Figure 2.** Global category map.

Table 4 summarizes the content of the units of meaning attributed in the content analysis of the interviews. After the detailed and systematic analysis of the data collected, it is confirmed that there is a clear approach on the part of the participants to the main themes that are associated with this research: Innovation, Procedures, Human Resources, Growth, Service Facilities, Balanced Scorecard, Intellectual Capital, Intangible Assets.

**Table 4.** Structure Units of meaning.

| Themes | Units of Meaning |
|---|---|
| Innovation | I01: "… whenever we work in a highly qualified market, innovation is central …" <br> I02: "… innovation and I believe that it is based on this that we are achieving the current prominence in terms of exports …" <br> I04: "… intellectual capital is measured through several reports …". |
| Procedures | I03: "… we naturally have regulations, however, we are not totally against excessive formalities …". <br> I04: "… we have procedures, however we are also given scope to contribute and have critical thinking …" |
| Human Resources | I01: "… all this success that we have been talking about has been achieved thanks to the effort of our human resources. Our ideas come from them and are developed by our resources …" <br> I05: "… Of course, cohesion is important … we usually spend more time in the front-office with the client or thinking about how to develop products for the client …" |
| Growth | I01: "… in the future there will be a need for specialized reports …" <br> I05: "… We are in a highly competitive market, so we really have to give importance to growth and the provision of new services …" |
| Service Facilities | I05: "… to be successful we need our customers to trust us and for that, we must work with greater commitment …" <br> I01: "… we really provide services and innovative products … I agree to say that we are oriented towards providing services …" |
| Balanced Scorecard | I02: "… we currently don't have the Balanced Scorecard implemented yet, but I think it would be a great asset …" <br> I04: "… Yes, in order to be able to measure the objectives and safeguard the link with the defined strategy, we consider that the BSC is effectively an excellent aid, as it allows to identify the cause–effect links between indicators and communicate the strategy to all stakeholders, whether internal or external to the organization …" <br> I05: "… Yes, the BSC is an excellent aid for linking the strategy to the indicators, setting goals and communicating the strategy itself. Additionally, it allows us to identify deviations from the defined strategy …" |
| Intellectual Capital | I03: "… Intellectual capital is behind each development, so when we try to determine the cost of a new product or service, we implicitly try to estimate its cost. However, it is not easy to add, as would be the case if palpable raw materials were involved …" <br> I05: "… we are considering the acquisition of an organization that competes with us, so we are fully aware that the value of the company does not correspond linearly to the value of financial reports …" |
| Intangible Assets | I02: "… from time to time, intangible assets will assume a prominent position … in the future there will be detailed reports about them …" <br> I01: "… it is really more difficult to be able to estimate the value of intangibles, however, it is this value that distinguishes us in the market" |

Regarding innovation, we found out during the interviews that innovation is indeed an important topic for the major innovative organizations in Portugal. Interviewee I said that innovation is central in highly qualified markets. Furthermore, there is a recognized link between innovation and prominence in terms of exports, which means that organizations that foster innovation promote their export visibility. Organizations that foster their differentiation, namely by adapting quickly to innovative requirements, will achieve a competitive and innovative position [37].

As innovation is directly linked to intellectual capital, it is known that there is still a need to have adequate reports to measure the grade of intellectual capital. This is especially relevant to

guide organizations that have opted to reduce their innovative assets, in order to try to estimate the impact. Usually, organizations that have decided to reduce their investments in innovation face a reduction in a short-term, but will face difficulties to recover their position due to uninterrupted market fluctuations [15].

Procedures are relevant to guide managers to follow tasks and understand the workflow. The interviewees mentioned that they understand the importance of having procedures, even if these sometimes cause implementation barriers and that they are not against formalities. In their eyes, procedures should exist, even if they consider that managers should not see the procedures in a limited view, but try to contribute with critical thinking.

As innovative organizations usually comprise several subunits, it is known that procedures make the understanding of the whole process, such as the segmentation of the market, easier [27].

Throughout the interviews, it was notable that success is directly linked to Human Resources. Interviewees are aware of the relevance that Human Resources assume for organizational performance. The managers of these major innovative organizations even go further, by saying that indeed Human Resources are the center of the success, as the ideas and development of new ideas are originated by Human Resources. In this vein, Learning and Development is the key level to leverage and promotes learning, adapting and growing. In order to ensure research and development, organizations have to efficiently allocate their collaborators [60]. Despite this importance—namely the correct allocation of collaborators—there is often a lack of time to develop products, as most of the managers spend too much time on activities related to the front office. Therefore, on one hand, Human Resources are valorized as being the main feature of organizational performance, but on the other hand, managers still have the perception that further relevance should be devoted to developing new innovative implementation ideas.

In the course of the interviews, it was found that the actual trend of growth will improve even more in the future, as nowadays, the need for growth is a feature that may compromise the longevity of an organization. Furthermore, in the interviews it was confirmed that, due to the very competitive market, the importance of growth and the implementation of new services will be of particular importance. In this vein, Learning and Development characterizes the learning and development of employees, as learning is considered to be a key for growth. Managers know that to forecast growth and adjust the development of new products, the development of Human Resources is relevant to enable this expected growth [3].

Regarding Service Facilities, it was stressed by the interviewees that they are very important, but strictly associated to these services is the trust that customers need to feel towards their supplier. Furthermore, interviewees confirmed that organizations devote special attention to providing services and innovative products, maintaining the orientation to provide adequate services. It is known that services have to meet customers' requirements, so competitive products and services need to be developed. Consequently, managers adapt organizational routines by configuring their activities towards customer's requirements [44].

Managers of the selected organizations declared that even if they do not have the BSC implemented, they do recognize the BSC as a great asset to improve and monitor intellectual capital. In addition, they state that with the BSC, they are able to make a link to strategy and understand the cause and effect relations of other indicators, such as their link to communication. They consider that the communication of strategies is crucial throughout the organization, which entails including all stakeholders, as communication is not only relevant inside the organization, but also outside, and relevant information needs to be provided to all stakeholders whether or not they are shareholders. Furthermore, in the interviews, another benefit of the BSC was highlighted, namely the possibility of tracking deviations, defining an action plan and setting corrective measures to prevent further and future deviations. Mainly, organizations have decided to implement the Balanced Scorecard to serve as a strategic management tool based on the need to overview performance [54]. Regarding these benefits, in the literature it has been pointed out that several managers have implemented the Balanced

Scorecard for several reasons, such as to monitor, budget and plan activities, or even to follow in an integrated way decisions related to investments [3].

Without any hesitation, interviewees highlighted that the source of any development is based on intellectual capital. Bearing this in mind, every time a new product is developed the cost of intellectual capital has to be considered and integrated. On the other hand, despite recognizing this relevance, interviewees know that it is difficult to measure the cost related to intellectual capital, as it is of intangible nature. Even so, managers know that when comparing with other organizations, the value that any company has is not just the value that most of the companies observe on Financial Statements.

It is known that intangible assets are very well regarded, as managers have understood in the recent past that by means of these valuable assets, organizations achieve great operational performance development. Despite having difficulty in measuring the value, individually, of intangible resources, there seems to be no doubt that intangible assets enable organizations to distinguish themselves. It might be stated that intangible assets leverage the differentiation of resources and consequently the positioning of organizations in a specific market. Over the past few years, the competitive advantage of intangible resources has achieved a great position, with them being considered unique resources. These resources are usually, or last least should preferably be, difficult to imitate. The innovative organizational models have contributed to sharing information, knowledge and organizational practices [9].

Regarding Innovation, finally, we highlight the strong relationship between Innovation, Intellectual Capital, Human Resources, Services Facilities, Results and Procedures.

For a better understanding of the study, Figure 3 allows us to understand the relationship between emerging themes and their saturation in the interviewees' discourse. It can be seen in this way that the main themes in the interviewees' discourse are: Intellectual capital, Innovation, Balanced Scorecard, Intangible assets, Human resources and Service facilities. The set of relationships shows that one of the themes studied—Innovation—is strongly associated with Intellectual capital, Balanced Scorecard and Intangible assets. This link suggests the need that companies feel to be increasingly innovative, whether through the acquisition of machinery, through process improvement or, increasingly present in the reality of companies, the need to professionalize their staff. Naturally, this need has a strong financial impact on the company, if the acquisition of machinery and process improvement implies a high financial effort.

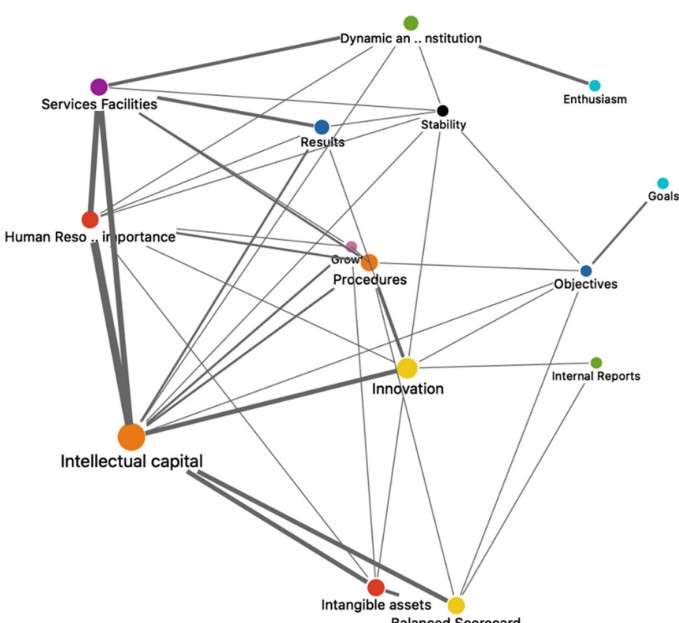

**Figure 3.** Occurrence and list of emerging themes.

The strong relationship between Intellectual capital and Human resources also corroborates the importance of employees and their knowledge for the success of organizations.

Likewise, we can see that one of the strongest relational triangles that we obtain is that involving the relationship between Intellectual Capital, Human Resources and Service Facilities that demonstrates that without knowledge, people and services, organizations cannot succeed.

Finally, we highlight the strong relationship between Innovation, Intellectual capital, Intangible assets and Balanced Scorecard, which are critical success factors for the management of a company.

In another perspective, Figure 4 helps us to understand, in a segmented way, the contribution of each participant to the identification of emerging themes, based on their reports. In terms of occurrence, the categories Intellectual Capital (25), Innovation (17), Intangible Assets (14), Service Facilities (14), Human Resources Importance (13), Procedures (12) Balanced Scorecard (11), are evidenced by the greater number of themes and sub-themes arising from their statements.

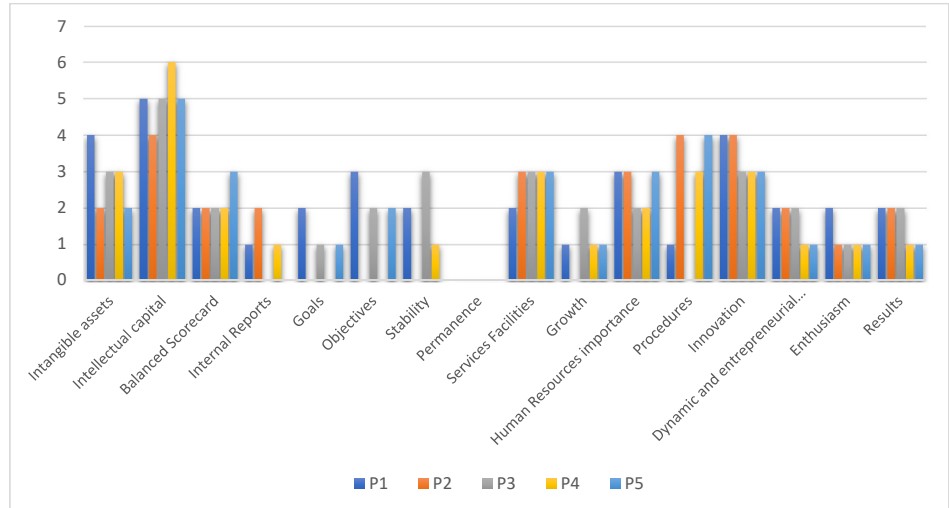

**Figure 4.** Occurrence of emerging themes.

Through the analysis software MAXQDA18, it was possible to build a word cloud (Figure 5) that coincides with a simpler lexical analysis, highlighting the most influential words in the corpus analysis of the interviews. Based on the image in question, we can observe the most frequent set of words in the speeches of the participants which are more associated with the problem under analysis, with greater emphasis on the terms: Intellectual Capital, Innovation, Intangible Assets, Service Facilities, Human Resources Importance. This output, namely Figure 5, reflects the keywords related to Innovation and its influence on tangible and intangible resource allocation.

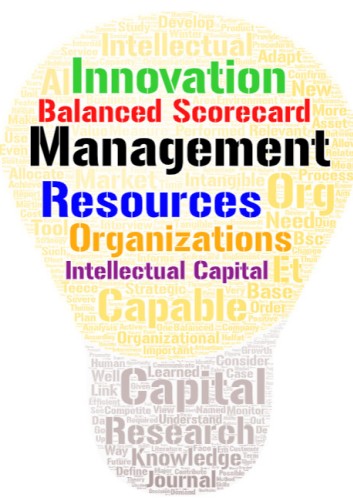

**Figure 5.** Word Cloud.

## 5. Conclusions

Bearing in mind the defined research objectives, namely to obtain more in-depth knowledge of the demands related to innovation and its consequent resource allocation, this study seeks to understand not only the influence of Innovation in tangible and intangible resource allocation but also the importance of performance measurement tools in monitoring tangible and intangible resource allocation.

This study sheds new light into this research area, as mainly the exploratory part of this research confirmed several statements in the recent literature. Furthermore, the interviews enabled us to confirm that the allocation of tangible and intangible resources, like Intellectual Capital, is crucial for the organization's position. Managers state that innovation is indeed a crucial topic, in order to be able to compete in very demanding and turbulent environments. In addition, this research confirmed the relevant interconnection of innovation and the export prominence of organizations, meaning that the grade of innovation might hinder, in case that innovation is not adequately explores, or on the other side expand export to highly qualified markets. Literature confirms that managers are able to enrich their distinction by adapting quickly to innovative requirements [37].

Having highlighted the relevance of resource adaptation, interviewees of the major innovative organizations confirmed the need to trust reports to guide managers. In this sense, as innovation is directly linked to intellectual capital, the reports have to fit managers' needs, namely to report adequately the grade of intellectual capital. Managers confirmed the importance of measuring intellectual capital, as several organizations lost control of intellectual capital, which consequently led to an unnoticed reduction in innovative assets. It is known that organizations that started to reduce their investments in innovation, have faced severe and sometimes unrecoverable consequences in market positioning [75].

Considering the relevance of market positioning and efficient resource allocation, a performance management tool was referred to as crucial, to guide managers. Nevertheless, besides the tool, there is a preliminary need to define and follow procedures. Managers of the major innovative organizations stressed the non-comprehension of the workflow as a barrier to performance tools in monitoring performance, due to the missing interlinkage to procedures. Interviewees confirmed that despite the fact that procedures sometimes behave as hinders to fast implementations, they should serve as a guide. Thus, managers should not see the procedures in a limited view, but try to contribute with critical thinking. As innovative organizations frequently comprise several subunits or processes, the need to have clear and efficient procedures is even higher to understand and interlink the whole process chain [27].

Furthermore, the interviews confirmed that managers consider that Human Resources have a direct link to organizational performance, as these are able to define the position of the organization in this very turbulent and competitive market. They even go further and confirm that success arises from Human Resources, as these resources are the basis for new ideas, new product developments or even improvement ideas. Consequently, the importance of learning and development is highlighted since it is based on the capability of learning, adapting and growing of resources that performance is promoted [60].

Referring to the need to have a management tool to guide in decision making, managers of the most innovative organizations confirmed that, independently of having or not having the BSC implemented, they consider the BSC as efficiently capable of guiding managers and improving and monitoring intellectual capital. Another relevant feature of the BSC consists in the possibility of tracking deviations and defining an action plan in order to set corrective measures. All in all, organizations chose the BSC as their performance measurement tool to have an overview of all the indicators [54].

This research stressed the importance of having deeper knowledge of tangible and intangible resource allocation and described the relevance of resources, particularly intangible resources, on organizational performance, thereby focusing on an existing gap in the literature.

Mainly for innovative organizations, the need to adapt and transform resources was highlighted, but despite the amount of research in this area, this has not been interlinked, as research seems to have

focused either on resources, intellectual capital or management tools in an isolated way. To the best of our knowledge, by interlinking the relevance of resource allocation, intellectual capital and their impact on innovation sustained by adequate measurement tools, this research indeed enriches the research area.

Regarding the global perspective of this research, it might be concluded that it adequately interlinks the related theoretic frameworks and enriches the field of empirical practices.

Managers will have a guide on understanding how to monitor resource allocation, independently of being tangible or intangible, based on a performance measurement tool.

Finally, it is expected that these findings and conclusions will contribute to a better understanding of innovation, consequent resource adaptation and its influence in intellectual capital and reporting, as a phenomenon of great social and economic relevance. This research field is a fertile one for future investigations, which may effectively and robustly enhance scientific knowledge.

## 6. Limitations and Suggestions for Future Research

Despite the selection of the case study method, which provided relevant in-depth information allowing for the collection of exhaustive information on the reality under analysis, its results and conclusions are limited to the selected major innovative organizations in Portugal, belonging to diverse activity sectors. For further investigations, multiple cases should be considered in several companies across the country, to allow for a comparative study.

Furthermore, a comparative study with other countries in order to compare and benchmark results of other economic and cultural contexts, would be enriching. The insights might be triangulated with quantitative approaches in order to outline new flows of further research.

**Author Contributions:** Investigation, R.S. and C.O.; Methodology, R.S.; Project Administration, R.S.; Software, R.S.; Validation, R.S.; Writing – Review and Editing, R.S.; Formal Analysis, C.O.; Visualization, C.O.; Writing original draft, C.O. All authors have read and agreed to the published version of the manuscript.

**Funding:** This paper is financed by National Funds of the FCT – Portuguese Foundation for Science and Technology within the project «UIDB/03182/2020.

**Acknowledgments:** The authors gratefully acknowledge the technical support received from NIPE (Centre for Research in Economics and Management), University of Minho and CETRAD (Centre for Transdisciplinary Development Studies), University of Trás-os-Montes e Alto Douro.

**Conflicts of Interest:** The authors declare no conflict of interests.

## Appendix A

### Interview Guide

---

*This Interview Guide comprises an international investigation into high-tech companies in Portugal.*

---

The data collected through this investigation will be processed using statistical tools and statistical tests. The search results will be presented in bulk and not individually.

Thank you for your cooperation!

SOCIODEMOGRAPHIC CHARACTERIZATION OF THE ORGANIZATION

1.  What is the legal form of the organization in which you work?
2.  Since how many years has the organization existed?
3.  What is the average number of employees in this organization?
4.  Has the organization had positive or negative results in the past year?
5.  Are accounting services provided internally or externally to the organization?

RESEARCH SPECIFIC QUESTIONS

6    Regarding the following statements, please state what you perceive about each one of them:

    6.1    I personally identify with this institution, which is an extension of my family. Do employees also share this feeling?

    6.2    This institution is dynamic and entrepreneurial. All are adventurous and defend the following proverb: "He who does not risk does not gain".

    6.3    This institution is oriented towards the provision of services. The biggest concern is to do the job with the greatest commitment of all.

    6.4    This institution attaches importance to human resources. High cohesion and morale are important in this institution.

    6.5    This institution attaches importance to the growth and provision of new services. It is important to be quick in the way you face

    6.6    This institution gives importance to competition and results. Achieving goals is important at this institution.

    6.7    This institution gives importance to competition and results. Achieving goals is important at this institution.

7    Regarding organizational objectives and goals, consider that these are monitored through a database.

8    The organizational objectives and strategies are reported only in internal reports.

9    Does the organization apply methods of measuring intellectual capital, not provided for in accounting standards?

10    Do you consider that the organization is characterized by high technological capacity and continuous investment in R&D solutions?

11    Are significant resources (hardware, software, people, among others) allocated to information technology and its users?

12    The real value of an organization is difficult to estimate, when it consists of human capital, because the knowledge and experience of employees is difficult to be quantified.

13    Do you agree that the use of standardized methods of evaluating intellectual capital allows a more realistic assessment of the value of a company?

14    Is the organization's management sufficiently familiar with the benefits and capabilities of the Balanced Scorecard (BSC) reporting?

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
