# Peer review of "The Influence of Innovation in Tangible and Intangible Resource Allocation: A Qualitative Multi Case Study"

_sustainability, doi:10.3390/su12124989_

Round 1

Reviewer 1 Report

There is not explanation of BSC acronym.

Also not stated from the title or abstract, the paper is quite heavy on balanced scorecard.

Please check if this is correct: "Consequently managers confirm a need to need to ensure a better control". (r.51)

"In this economic and social context, decisions must be taken quickly, knowing that if the decision is postponed, certain opportunities may have already been missed (Eisenhardt, 1989)". While I believe the assertion to be correct, a reference from 1898 made as "THIS economic and social context" appears dated. We are so far from the '89 context...

The literature review is quite lengthy, yet dated. There is a predilection of using sources from the '90s and early 00's.

The number of interviews appears to be so small...

Author Response

Dear Reviewer, thanks for your great job and enforces to give me ideas and good advices to improve the paper.

We made numerous changes to the paper, in several sections. Your calls for attention were very important to improve the article.

See please, the changes in green.

We change the manuscript organization like you proposed;

Reviewer 2 Report

The study aim is to explore the influence of innovation in tangible and intangible resource allocation. Especially the intangible resources like learning and intellectual capital are very welcome in the modern organizations.
This issue is rather interesting for readers, although the research are not presented in a quite clear way.
I presented my specific comments below.
The research explores the field of resource allocation, namely dynamic capabilities and highlights the importance of monitoring intangible resources. The scientific hypothesis should be clearly stated in the abstract / introduction section. The authors should also clearly explain what they mean as “the influence of innovation” in the relation with the paper topic.
In the theoretical section the authors analyse some interesting topics, but the sections 2.1 – 2.4. should be enriched by the short conclusions presenting the relevance of each topic to the research problem of the paper.
Regarding the section - 3.1. Methods as far as I understand the authors approach, I see that they focused on one research method – case study supported by the interviews with 5 respondents. I think it should be much better to describe briefly the cases which were analysed by the authors.
Regarding the research method – I would like to ask authors to explain why they analysed the correlation between the units of meaning and the emerging themes but only for 5 interviews. In my opinion this is not a representative sample. I think that authors should better explain why they chose exactly this approach and present the wider range of other possibilities to solve the research problem. For example they should mentioned other research regarding the similar topics and their results. It should be also analysed in the net section – discussion.
The section 4. Results and Discussion should be supported mainly by the authors research results but it is based mainly on literature review. It cause some doubts if the research result are enough trusty worth to allow the authors to define conclusions. It should be explain how the authors defines the Figure 2 - Occurrence and list of emerging themes – analysing only 5 interviews. The authors should convince the readers that sample 5 interviews it is a reasonable sample for such research.
Generally the issue is interesting but needs to be corrected at least according to the above remarks.
Good luck!

Author Response

(The authors gave the same response as above.)

Reviewer 3 Report

The manuscript entitled “The influence of Innovation in tangible and intangible resource allocation: a qualitative multi case study” is quite interesting and informative to most readers of this field. 

However, the following issues should be addressed before considering the manuscript for publication:

  • The abstract is too long.
  • Don't use acronyms (BSC) without introduction.
  • The research question must be better contextualized and be more convincing. I suggest, that the authors give more introduction about why your study is important in the section of introduction.
  • The structure (outline) of the paper could be given at the end of the introductory section-
  • The literatures review must be better contextualized and be more convincing. In order to be valid, this manuscript must include a proper analysis of the relevant literature and then make a comparison with the authors' approach. For instance, the authors should organize all the reviewed papers in a table and compare the difference among these reviewed papers and this study. Furthermore, I think that there is sometimes a lack of focus as there are too many concepts that are loosely connected between them.
  • Research methodology is not very clear. I suggest that the authors use a flowchart.
  • Figure captions should be below the figures; table heads should appear above the tables.
  • Appendix 1 is missing. More details about the questionnaire should be provided.
  • Is there any other source of information to support the so-called expert opinion? At least a discussion on the validity of expert opinion be provided.
  • Figure 2 deserves much more in-depth discussion.
  • Figure 4 added nothing for me.
  • The authors should convince the readers of Sustainability, that their contribution is so important. The main contribution of this manuscript should be compared with other similar empirical studies.
  • What are the managerial implications from this research?
  • How decision or policy makers could benefit from this study.
  • Finally, I am still not convinced, for the scientific value of the paper. It would increase the impact of the manuscript, If the authors, try to indicate, this explicitly in the manuscript.
  • As usual a final thorough proof-reading is recommended.

Author Response

(The authors gave the same response as above.)

Round 2

Reviewer 3 Report

The paper has significantly improved as compared to the previous version. I am also quite happy with the responses and explanations given by the authors to my comments.

Thus, in my opinion the paper is recommendable for publication.